# Trends and Limits for Quinoa Production and Promotion in Pakistan

**DOI:** 10.3390/plants11121603

**Published:** 2022-06-18

**Authors:** Irfan Afzal, Shahzad Maqsood Ahmed Basra, Hafeez Ur Rehman, Shahid Iqbal, Didier Bazile

**Affiliations:** 1Department of Agronomy, University of Agriculture, Faisalabad 38040, Pakistan; shehzadcp@gmail.com (S.M.A.B.); hafeezcp@gmail.com (H.U.R.); 2Department of Agronomy, MNS-University of Agriculture, Multan 66000, Pakistan; shahid.iqbal@mnsuam.edu.pk; 3CIRAD, UMR SENS, 34398 Montpellier, France; 4SENS, CIRAD, IRD, University Paul Valery Montpellier 3, 34090 Montpellier, France

**Keywords:** Andean regions, abiotic stresses, nutrition profile, value chain, developing countries, germplasm diversity

## Abstract

Quinoa is known as a super food due to its extraordinary nutritional qualities and has the potential to ensure future global food and nutritional security. As a model plant with halophytic behavior, quinoa has potential to meet the challenges of climate change and salinization due to its capabilities for survival in harsh climatic conditions. The quinoa crop has received worldwide attention due to its adoption and production expanded in countries out of the native Andean region. Quinoa was introduced to Pakistan in 2009 and it is still a new crop in Pakistan. The first quinoa variety was registered in 2019, then afterward, its cultivation started on a larger scale. Weed pressure, terminal heat stress, stem lodging, bold grain size, and an unstructured market are the major challenges in the production and promotion of the crop. The potential of superior features of quinoa has not been fully explored and utilized. Hence, there is a need to acquire more diverse quinoa germplasm and to establish a strong breeding program to develop new lines with higher productivity and improved crop features for the Pakistan market. Mechanized production, processing practices, and a structured market are needed for further scaling of quinoa production in Pakistan. To achieve these objectives, there is a dire need to create an enabling environment for quinoa production and promotion through the involvement of policymakers, research institutions, farmers associations, and the private sector.

## 1. Introduction

Climate change, water shortage, and increasing salinization including malnourishment and chronic dietary problems are the major challenges for sustainable agriculture as well as for food and nutritional security of the burgeoning population. It is the right time to diversify cropping systems by introducing new crops to achieve sustainable development goals [1]. Quinoa is an ideal candidate crop which may contribute to environmental and food sustainability owing to its high adaptability to a wide range of growing conditions [2]. Quinoa is gaining popularity due to its functional and nutritional characteristics [3]. It can achieve higher productivity and maintain nutritional quality in different environments where conventional crops cannot perform well. Moreover, quinoa has potential for climate resistance to different stresses such as salinity, drought, and frostlike conditions [4,5,6]. It is an annual, mainly self-pollinated, dicotyledonous, and C_3_ crop for CO_2_ fixation during photosynthesis [4].

Quinoa has a high nutritional profile with 10–18% seed proteins [7,8] and 4.1–8.8% fats [9]. It is ideal for celiac patients because it is gluten free. The whole plant can be used as feed for both humans and animals. Its leaves are also used as a salad because they have the same nutritional value as spinach and mustard [10]. Quinoa grain is rich in all amino-acids, vitamins (A, E, B2), carbohydrates, minerals (K, Fe, Ca, Mn), and healthy supportive fatty acids (Omega-3) [9]. Its grains are ground into flour as wheat and used for further purposes such as bread formation, beer formation, and fermented drinks [11].

Quinoa has been cultivated in more than 120 countries worldwide with major producers including Peru, Bolivia, Ecuador, USA, Columbia, Chile, and Brazil [12]. 

The quinoa plant life cycle is divided into vegetative and reproductive stages. Each phase is dependent on day length and temperature [13] due to which it has wide adaptability [9]. Quinoa plants take about 40–89 days for bud appearance, 7–50 days for the anthesis stage, and 66–135 days for maturity after anthesis [14]. However, the crop reaches maturity within 109–182 days in Europe [15,16]. 

In Pakistan, quinoa was introduced for the first time in central Punjab by the University of Agriculture, Faisalabad (UAF), to increase diversity in the cropping system and environmental sustainability [17,18]. Now it is well adapted and grown in all provinces of Pakistan.

Over 7 million hectares in Pakistan are affected by soil salinity. Research indicates that quinoa can be grown on salt affected soils with electrical conductivity (ECe) 10 to 15 dS cm^−1^ in South Punjab [19,20]. It can even tolerate salinity and arsenic stress due to less uptake of toxic ions and higher activities of antioxidant enzymes [21]. Therefore, quinoa crop has potential for salt affected soils. Despite huge potential and wide adaptability, lack of awareness about nutritional and health benefits and unstructured markets are major challenges in upscaling quinoa crop in Pakistan. This review highlights the current trends in quinoa research, its cultivation and future challenges in quinoa production, and value chain development in Pakistan.

## 2. Germplasm Collection and Evaluation

Only a few quinoa varieties have been commercialized out of more than 3000 landraces identified in the Andean countries [22]. Cultivated quinoa has plentiful seed colors (>10), but the marketable grain is usually white, red, and black. During 2009, quinoa was introduced successfully in Pakistan based on a collection of 170 accessions from the USDA, USA, and Denmark [17]. Out of 170 quinoa lines tested, only four accessions were found to be widely adapted to the local climatic conditions of Pakistan and valuable for domestic production. Basic farming practices have been developed by optimizing sowing time; sowing method; and nitrogen, phosphorus, and potassium (75:60:50 kg ha^−1^) requirements under Faisalabad conditions. The preliminary trials have shown that quinoa is also well acclimatized to the different agro-ecological conditions of Punjab. Yields obtained (3.2 tonnes/ha) and nutritional profiles investigated in these environments are equivalent to native regions of quinoa production [16]. Likely, adaptability trials across different parts of the country including KPK and Sindh are in progress. 

The University of Agriculture, Faisalabad, has conducted trials on genetic variability for a wide range of quinoa types under agro-climatic conditions of Faisalabad, Punjab-Pakistan, in collaboration with King Abdullah University of Science and Technology, Saudi Arabia. About 370 accessions have been phenotyped for morphological, phenological, and yield traits under field and for postharvest management of quinoa seed during the years 2019–2021 [unpublished data]. 

The UAF-Q7 is the first approved variety of quinoa in Pakistan [23]. The basic production practices for this variety have been optimized [17]. The UAF-Q7 variety has a hollow stem with a tap root system, and its leaf shape resembles the goose foot type. Its plant height ranges from 110 to 150 cm. The panicle shape is an intermediate type with a green color at flowering that turns brown at maturity. It matures in 130–140 days and has an average yield potential of 3.2 t ha^−1^.

## 3. Developments in Quinoa Research

Due to high grain yield, biomass, and nutritional quality, quinoa is regarded as a dual-purpose crop both for grain production and livestock feed [24]. After seed harvesting, there is potential for quinoa growers to market straw a forage crop [24,25,26]. Generally, genotypes with thick stems, more branches, and moderate plant height producing higher biomass are ideal for fodder and genotypes with compact inflorescence are ideal for grain purpose [27]. Presently, some quinoa lines with high nutritional profile, biomass, and low saponin contents have been evaluated for fodder purposes [unpublished data]. The assessment of the digestibility and palatability potential of these quinoa lines for ruminants is in progress.

### 3.1. Germplasm Diversity

Quinoa is cultivated from sea level to 4000 m with a broad agroecological adaptation to different types of soils [28]. It is an Andean crop that originated around Lake Titicaca in Peru and Bolivia, the area with greatest diversity and genetic variation. Currently, quinoa is grown in countries spanning five continents, including North America, Europe, Asia, Africa, and Oceania. The center of quinoa diversity is the southern Andean highlands viz. Bolivia and Peru have huge variability and Bolivia’s gene bank center has more than 5000 accessions [29]. Quinoa varieties are genetically grouped into two main groups: lowland and highland. Fuentes et al. [30] mapped quinoa’s genetic structure by matching it with natural geographical edaphic climatic constraints and the social linguistic context of ancient people inhabiting the Andes region [30]. 

Huge diversity also exists in the quinoa germplasm based on morphological and physiological adaptability to various climatic conditions [31]. Yield mainly depends upon the phenological and seed related attributes of a crop and duration between each stage. Under local conditions of Faisalabad, Pakistan, exotic accessions with medium crop duration and more plant biomass produced higher grain yield as compared to long duration genotypes. The number of lateral branches in quinoa plants vary according to the genotypes and the crop condition. Accessions with more branches and inflorescence express more plant biomass and yields as compared to accessions with a single panicle per plant [16]. 

According to Sosa-Zuniga et al. [32], 15 panicle colors and 3 types of panicle shapes (Glomerulate, intermediate, and amaranthiform) are reported in quinoa at physiological maturity. The large grain size in quinoa is preferred [3]. Apart from phenological and grain characteristics, quinoa genotypes also diversified in terms of nutritional quality as protein contents ranged from 11 to 16% in selected genotypes adapted in Pakistan [33]. Fewer studies have reported on the role of phytates in quinoa as it is known as an anti-nutritional factor. 

### 3.2. Production Practices

Quinoa can grow in a range of soils from clayey to sandy including marginal soils with a pH of 4.5–9.5 [13]. For quinoa cultivation in a new environment, sandy loam soils with good drainage, appreciable organic matter, and nutrients should be preferred. In Pakistan, quinoa crop has been preliminarily tested on sandy loam and clay loam soils with a pH range of 7.4–8.8, medium in fertility and low in organic matter contents (0.77%) under semi-arid regions of Punjab (elevation 184 m above sea level 31.4187° N, 73.0791° E; elevation 190 m above sea level 31.8950° N, 73.2706° E) [17,34,35] and Sindh (20 m above sea level) [36]. Quinoa is grown during rabi season as a spring crop in most parts of the country except for northern areas. The window of plasticity for planting ranges from 15 October to 15 December and favorable time for its growth and yield potential is during November under irrigated conditions [17,34,36]. A delay in planting the crop usually prolongs growth, reduces grain filling, and delays crop maturity with a substantial reduction in seed yield [37] and response may be genotype specific [38,39,40,41]. 

The crop sowing requires fine textured, well drained, and levelled seedbed with optimal moisture for its germination; but it is important to know that quinoa is sensitive to high moisture due to its small seed [4]. The planting method and geometry are critical in crop establishment of quinoa because of slow growth rate until the bud formation stage, otherwise weed-crop competition becomes greater to affect yield. Timely sowing of quinoa can provide a head start over weeds as crop may obtain good growth during this period. 

Experimentally, quinoa has been cultivated on ridges manually or by hand drill on normal and salt affected soils to sowing depth of 2–3 cm at field capacity level [17,34]. Ridge cultivation is usually practiced by many growers with plant distance of 15 cm on 75 cm spaced ridges [17]. Weeds are cumbersome to control; hence optimum plant density is important to reduce weed competition. Experimentation is in progress by planting quinoa at 30 cm inter-row distance and a plant distance of 11 cm using the drill method. Seed rate in quinoa depends on the method of sowing viz. 5–7 kg ha^−1^ for the drill method and 4–5 kg ha^−1^ for ridge cultivation. Nonetheless, high biomass, growth, and yield have also been reported in quinoa sown on beds with 75 cm width and 15 cm plant distance of a furrow on both sides for water flow under irrigated conditions [35]. This method has an advantage of planting quinoa on both sides of the beds compared to ridge planting with a single row [35]. Though, further studies on resource use efficiency in terms of water, fertilizer, and radiation including stem breaking under high wind and thunderstorms are required. 

Quinoa is produced in marginal lands of its native regions. Although, the crop is very fertilized and irrigation input is responsive under irrigated conditions. In Pakistan, a recommendation for nitrogen (N), phosphorus (P), and potassium (K) (N:P:K) using 75:60:50 kg ha^−1^ for quinoa cultivation is being followed. A full dose of the phosphorus and potassium and 1/2 dose of nitrogen are applied as basal and the remaining at the flowering stage [17,34]. Usually, high N application has been reported to delay maturity, increase plant height, and the crop may be susceptible to lodging [17]. Alandia et al. [42] discovered that increase in N rate 80–160 kg ha^−1^ resulted in a 10–15% rise in seed yield, while enhancing N rate up to 240 kg ha^−1^ resulted in negligible seed output. Furthermore, extensive research concerning nutrition in relation to soil type should be conducted before recommending farming practices for any specific location. As Pakistan soils are of alkaline nature and low in organic matter and micronutrients (Z, Fe, Mn, B), these essential micronutrients should be included in the basic fertility plan to harvest high quality quinoa grains. 

Quinoa is a drought-tolerant crop and has a low water requirement, though yield is significantly affected by irrigation [2]. Between three and four irrigations are required by a quinoa crop during its growing cycle; however, crop stages critical for its irrigation during the vegetative and grain formation period remain to define for its successful adaptation to semi-arid condition of country. Heavy watering throughout the panicle development phases has been reported to extend crop maturity and increase plant height, suggesting that the crop might be prone to lodging [Personal observation]. 

Various narrow and broad leaf weeds occur in quinoa fields and are mainly influenced by the type of sowing method, planting geometry, and plant density. Quinoa plants resemble its wild relatives *C. album* and *C. murale*, during the early growth period. Therefore, quinoa seedlings must be differentiated for proper identification of weeds and their control. As there is no chemical weed control yet established due to sensitivity of Chenopodium to herbicides, weeds are controlled manually. In research trials, weeds are controlled usually at 2–4 true leaf and bud formation stages to achieve optimum plant density [17,35]. Studies are much needed to establish the critical crop weed competition period in quinoa and combined application of different pre- and post-emergence herbicide formulations without detrimental effects on soil and plant foliage including their residual effects on the environment. Nonetheless, an integrated approach which involves mechanical, cultural, chemical, and biological control is called sustainable weed management in quinoa organic production.

### 3.3. Abiotic Stresses

Abiotic stresses are becoming the most devastating threat that limits agricultural productivity for most of the crops [43]. One of the possible solutions to cope with abiotic stresses is the cultivation of stress resistant crops to abridge the food requirement [44]. Quinoa is cultivated due to its abiotic stress tolerance behavior [45]. Due to this potential and unique nutritional profile, FAO termed quinoa as “Future Smart Food” and advocated for its promotion, especially in salt affected and drought prone areas [22,44]. 

Quinoa genotypes well adapted to local conditions had been evaluated for salt tolerance, heat, and phytoremediation potential [34,46].

Quinoa has been identified as a facultative halophyte with better salt tolerance [47] and a high variability in salinity tolerance among quinoa genotypes has been reported [48,49,50]. Saleem et al. [51] investigated the salt tolerance behavior of various quinoa lines grown hydroponically at 100 mM NaCl salinity level and found that Q7 and Q9 lines had better chlorophyll content index, free proline, ascorbic acid, and carotenoids contents but gaseous exchange traits decreased in Q7 plants under saline environment. In another study, Iqbal et al. [34] found an improvement in water relations, leaf photosynthetic rate, K^+^ contents in leaf, proline, phenolics, morphological and yield related attributes and ultimately increased grain yield at 10 dS m^−1^. Quinoa performance decreased drastically at 30 dS m^−1^. Iqbal et al. [19] also found that under natural salt affected conditions (9.8 and 13.9 dS m^−1^), leaf antioxidants, K^+^, total phenolics, and proline contents increased compared to control conditions while 1000-seed weight, grain protein, Cu^+2^, Ca^+2^, and Zn^+2^ contents were not affected [19]. However, seed and biological yields diminished under high salinity (>13.9 dS m^−1^) might be ascribed to poor seedling emergence caused by dispersion effects in sodic soil [52]. Yet, seed yield reported by Iqbal et al. [19] was higher (≈1 t ha^−1^) than world average yield under salt affected conditions [3]. Abbas et al. [20] reported that quinoa significantly improved plant biomass, grain number and weight, antioxidants, total chlorophyll, and relative water contents at 10.5 dS m^−1^. 

High temperature is one of the limitations to widespread cultivation of quinoa. Under the climate change scenario, the high temperature causes drastic effects on plant functions [53]. Rashid et al. [54] reported that quinoa plants under terminal heat stress induced 76 days after sowing produced less chlorophyll contents and decreased gaseous exchange parameters, seed yield and its nutrients. Contrastingly, plant height, antioxidants, seed Mg^+2^, K^+^, and Na^+^ contents were increased in heated plants as compared to controlled conditions. In another study, Rashid et al. [55] observed lower gaseous exchange, panicle length, 1000 seed weight, seed yield, seed Ca^+2^, K^+^, and chlorophyll content during anthesis when exposed to terminal heat stress. Control quinoa seeds, on the other hand, showed more antioxidant enzymes activity [54,55]. Quinoa performance was negatively affected when it was planted late in Pakistan conditions. At temperatures above 35 °C, quinoa performance suffers due to phenological changes which promote more vegetative growth than reproductive growth [3,17]. Quinoa is a cool season crop and sensitive to high temperature stress for grain production.

Heavy metal toxicity hinders the physiological, biochemical, and morphological responses which ultimately limits the yield of crops [56]. On the other hand, tolerance and plasticity in quinoa against heavy metals have been reported [57,58,59]. Parvez et al. [21] reported that at 150 μM arsenic (As) stress, seedling biomass, and chlorophyll contents were decreased while antioxidant enzymes increased. Under lead (Pb) 100 mg kg^−1^ and 60 mg kg^−1^ cadmium (Cd) stress, quinoa seedling biomass, and membrane stability index decreased, while tissue Pb, Cd, and antioxidant enzymes were increased [60]. 

Recently, Naeem et al. [61] found a decrease in seedling vigor and membrane stability index and a concomitant increase in root/shoot growth, SOD, POD and CAT activity including grain Cd contents at 75 mg kg^−1^ cadmium (Cd) stress. Haseeb et al. [46] found a decrease in morphological, yield related attributes and final grain yield and an increase in soluble phenolics, root, stem, leaf, and seed Pb contents at 100 mg kg^−1^ Pb stress. More importantly, Pb contents in quinoa grain were within the permissible limits (0.3 mg kg^−1^ DW) as per FAO/WHO guidelines [61]. This depicts the phyto-extraction capacity of quinoa against industrial effluents, mainly heavy metals. 

### 3.4. Phenotyping Approaches

Phenotyping is a foundation of plant breeding and grain yield is the most reliable phenotypic trait in the breeding programs [62]. Secondary traits also contribute to crop improvement depending upon genotype by environment interaction under various environmental conditions [63]. Three main classes of phenotyping are identified in the literature: handy, high-throughput, and precision phenotyping traits to tackle current bottlenecks to yield improvement [64]. Many useful phenotypes were established with the publication of *Descriptors for Quinoa and Wild Relatives* [64] and the guidelines for distinctness, uniformity, and stability testing of quinoa under CPVO system [65]. However, a detailed explanation of the important traits was lacking and needs further investigation. Regarding precision and high-throughput phenotyping through remote sensing, no work has been reported on quinoa in Pakistan. Studies for a consensus on phenotyping methods for 400 quinoa accessions in the field with the international collaboration are in progress, during which the phenotyping protocols at different phenological stages, maturity time, harvest and postharvest phases throughout the growing season have been established. 

Quinoa genotypes show different behavior in phenological stages and duration to complete their lifecycle according to the latitude, altitude, and environmental conditions especially photoperiod and temperature of a region [65]. Sosa-Zuniga et al. [32] presented the most recent and comprehensive description of phenological stages of quinoa in accordance with the BBCH criteria. For reliable and stable phenotyping, defined phenological phases are critical. Researchers defined eight major phases of quinoa crop development. However, stage five, inflorescence, is the crucial bordering phase between vegetative and reproductive growth stages. Additionally, stage six, flowering, is highly associated with yield related traits. Moreover, sowing and harvest dates are also important to record according to the local conditions. 

The duration of each quinoa stage is highly dependent upon temperature and photoperiod which is different for each quinoa variety [13]. In Pakistani conditions, exotic quinoa accessions along with UAF-Q7 reached the inflorescence emergence stage within 45–71 days after sowing and completed anthesis at 70–108 days after sowing. The accessions having emergence with UAF-Q7 completed the physiological maturity stage within 101–144 days after sowing (unpublished data; Table 1). In South America, days to flowering varies from 71 to 101, days to maturity varies from 117 to 157 days after emergence and seed yield (t/ha) varies from 0.32 to 9.33 [66]. In European region conditions, the total growth duration of quinoa crop varies from 109 to 182 days. In England, the appearance of true leaves to the visible floral bud initiation stage varies from 41 to 89 days, the visible floral bud stage to anthesis stage ranges from 7 to 53 days and maturity is reached from 65 to 135 days after anthesis [67]. 

### 3.5. Grain Nutritional Profile

Pakistan has the world’s sixth highest population by human index, which has a drastic impact on the world food program. A decrease in food security and safety has led to a child stunting rate of 45% in Pakistan, ranking 8th among 132 nations [68]. Such conditions increase the healthcare costs of this lower-middle-income country. Quinoa’s diverse nutritional profile can offset prevalent nutrient deficiencies related to the lack of nutrient-dense or biofortified crops. Nasir et al. [33] investigated the nutritional profiles of grains obtained from Pakistan’s well adapted quinoa genotypes (Q1, Q2, Q7, and Q9). Genotypes of quinoa were evaluated with special emphasis on functional properties and digestibility of its proteins. Proteins of all genotypes had good functional properties, i.e., water absorption capacity (2.81–3.82%), oil absorption capacity (2.72–3.03%), and foaming capacity (9.09–10.05%). Proteins also exhibited outstanding in vitro digestibility (75.95–78.11%), protein efficiency ratio (3.5–3.78%), net protein ratio (3.9–4.69%), net protein utilization (70.75–73.78%), biological value (79.15–81.74%), and true digestibility (87.66–90.57%). Fats were also studied, and various fatty acids were found including oleic acid (26.28–31.62%), palmitic acid (11.39–13.25%), α-Linoleic acid (4.45–7.71%), and Linoleic acid (47.73–52.02%).

Iqbal et al. [19] estimated the nutritional profile of quinoa grains obtained from crops grown on fertile and salt affected soils. Highly significant results showed the resilient nutritional profile of quinoa grains via depicting no change in the quality of grain protein contents. Astonishingly, seeds harvested from salt-affected soils were rich in potassium, magnesium, and manganese. Mineral profiles of quinoa grains adapted to Pakistani soils are given in Table 2 and Table 3.

Vega-Gálvez et al. [73] studied detailed characterization of the nutritional composition of six quinoa varieties grown in Southern Europe. High contents of potassium, phosphorus, and magnesium along with low saponin contents were reported in these quinoa varieties. Nonetheless, further studies are required to explore amino acid profile, antioxidants, and identification of bioactive compounds such as kaempferol and quercetin.

### 3.6. Seed Storage

Quinoa seed quality depends on environmental conditions at the time of harvesting and storage [74]. Proper handling and safe storage ensure seed quality at the time of sowing. Temperature, moisture contents, and oxygen are important factors that influence seed longevity [75] but elevated seed moisture is the most critical factor responsible for loss of seed quality during storage [76,77]. Poor storage enhances the attack of storage insect pests, which promotes deterioration, and eventually death of seeds [78]. Due to inadequate storage, both natural and economic resources are spoiled if poor quality seeds are sown in the field [79]. So, the quality of seed should be maintained during production, harvesting and storage to ensure the availability of highly viable seed at the time of planting. 

Quinoa seed is spherical and consists of a peripherally curved embryo surrounded by a large central perisperm, a two-layered pericarp, and a seed coat. A micropylar endosperm in the form of a cone surrounds the radicle tip [80]. Quinoa seeds lose viability more rapidly than cereals because of the porosity in the integument, which allows a seed to easily gain or lose moisture and may initiate germination in the panicle [74]. Initial quality of seed, temperature, and humidity during storage and rate of aging process influence seed longevity [81]. This aging process varies among quinoa accessions [82]. Quinoa seed deteriorates with an inadequate storage environment, particularly at high relative humidity and temperature [74]. Recently, Kibar et al. [83] reported loss in viability of quinoa seed packed in traditional bags during storage at ambient conditions. Conversely, if a seed is dried properly and packed in hermetically sealed storage bags, the quality of quinoa seed could be maintained as reported in other cereals [76]. If seed loses its viability under ambient storage conditions, then it would be very difficult to obtain optimum plant population in the field as quinoa seed is very sensitive at the seedling development stage. Furthermore, environmental factors such as high temperature and moisture during production can also influence seed quality of quinoa. 

During storage, seed moisture contents, relative humidity, and storage temperature are the main factors that determine the viability of quinoa and rate of deterioration [84,85]. Dry storage of seed for a short-term period preserves its biological value. For long-term and reliable storage, specific cold storage conditions have been used [86]. Despite that, seeds still deteriorate at a reduced rate in the dry state due to very low levels of metabolism [87,88]. Decline in seed quality is initially seen as a decrease in rapidity and synchronicity of germination. An increasing delay to germination is also accompanied by an increased frequency of abnormal seedlings in quinoa seeds and eventually demonstrates a loss of viability. Quinoa seed is orthodox and hygroscopic in nature so it can gain moisture from atmosphere and become susceptible in storage. Seed moisture determines the total life span of vigorous seed so drying is performed after harvesting for reducing moisture contents and to increase storage duration. For quinoa, approximately 10% moisture contents are best for prolonged storage [89]. At 18–20% moisture content and 70–80% RH, the respiration rate increases, and metabolic reactions start. Temperature increases the rate of deterioration in the presence of moisture contents and humidity. High temperature along with high moisture content promotes dormancy as well as ageing in quinoa.

## 4. Quinoa Consumption and Product Development

The grain composition of quinoa shows health benefits concerning contents of fatty acids, minerals, good quality protein, and bioactive compounds. For these reasons, its consumption is adopted by health-conscious citizens. Quinoa is consumed as a significant ingredient in meatballs and salads and is used to prepare cookies as gluten-free products. Several other products that include quinoa ingredients are multigrain flour such as Maxgrain product to supplement nutrition for people consuming monotonous single grain flour especially for diabetic and celiac patients. A recent development is the launch of CERELAC with oat and quinoa by Nestle-Pakistan for nourishing infants and children. Several other local quinoa-based recipes are being branded to increase quinoa consumption as “Kheer”, milkshakes, fruit salads, chapati, kebabs, and vegetable salads. Quinoa may be added to bread flour after evaluating the functional properties and digestibility of protein of available quinoa genotypes [33]. In another study, Mahmood et al. [90] evaluated the rheological properties of quinoa, buckwheat, and wheat doughs and sensory properties of cookies made from their flours. They found good nutritional benefits and high sensory acceptability from composite flour having 10% quinoa and 10% buckwheat. It is proven that quinoa genotypes grown in Pakistan have a strong nutritional profile, especially better protein quality [33,90]. Thus, it can be utilized in cereal-based products for achieving higher quality and value addition. Almost 21 food companies have introduced quinoa products in the country and most of these companies are also involved in the export of quinoa in UAE and European countries (Table 4).

## 5. Challenges in Quinoa Production and Promotion

Quinoa expansion and production started across the globe after its recognition by the United Nations in 2013 [22,91]. During the year 2007–2008, it was agreed in West France to grow quinoa “d’Anjou” in the Loire area. Trials in Italy indicated that quinoa can be grown in southern regions, and it thrives even in harsh natural conditions. Positive studies have also been performed in Morocco, Greece, and the Indian Subcontinent (India and Pakistan). India is particularly interested in establishing its own quinoa markets [3].

A common issue in these countries is a system with a small or non-existent market, farmers who are risk averse and severe lack of information and technical diffusion. Finally, there is a growing trend to test this crop under local conditions to expand national markets. As a result of the rise in demand, price hikes were observed which have tripled between 2006 and 2013 [3]. Carimentrand et al. [92] concentrated on the various approaches of selling mixed quinoa grain in local markets by responding to international and domestic demand for standardized quinoa products. Agro-industrial enterprises and exporters have encouraged farmers in Peru and Bolivia to plant improved quinoa varieties to meet market demand for uniform and large grains. Community resilience and socio-economic challenges of the quinoa market must be taken into account concerning environmental challenges in quinoa value chain [93]. It is emphasized that rising worldwide market prices have resulted in a drop in consumption, specifically in quinoa growing regions [94]. 

Despite the growing worldwide recognition of the health benefits of quinoa, the barriers to its widespread adoption remain significant. Institutions and farmers are facing a lack of knack in terms of planting, harvesting, distribution, and overall management. Furthermore, rural residents are unaware of the crop’s nutritional benefits; they are not used to the taste and lack recipes to incorporate this product into traditional dishes for consumption [95]. Lack of factors such as information, training change in agronomic, and plant protection practices are the constraints in adoption of quinoa cultivation [96].

### 5.1. Mechanization

For sustainable cultivation of quinoa crop, proper management of chemical fertilizers and farm machinery are the key factors [97]. Additional characteristics include modifying land use and mechanization of agricultural practices [92]. When compared to a manual production system under rain fed, using mechanized production and processing practices combined with irrigation and organic amendment can reduce processing costs from 2.8 to 1.2 USD kg^−1^ [98].

Sowing methods have a great influence on growth, morphology, yield, and biomass accumulation. The raised bed planting technique is superior for obtaining high grain yield under the irrigated conditions of Pakistan [35]. Quinoa seed sowing by hand is being practiced in developing countries such as Pakistan which is labor intensive and high seed rate demanding. Similarly, harvesting is also performed by hands so mechanization at sowing and harvesting times is a big challenge for the quinoa growers in the developing countries. 

The industrial processing of quinoa is crucial to ensure the consumer or supplier is provided with clean quinoa, free of impurities and saponins. Since 2009, quinoa was introduced in Pakistan, but its cultivation is limited because of bitterness in approved varieties which is attributed to its high saponin contents. Farmers are practicing a traditional method of washing and drying for its removal which is a labor-intensive process. The timely introduction of mechanized system at harvest and postharvest stages has various advantages over traditional practices. In Morocco, mechanical pearling, on the other hand decreased saponin content by 68%, compared to 57% using both conventional abrasion and cleaning [98].

### 5.2. Weed Control

Weed control is an important crop husbandry practice since quinoa grows in a season when its wild relatives, such as *Chenopodium album* and *Chenopodium murale*, compete for light, water, nutrients, and space. It is difficult for common farmers to distinguish among all these at early growth stages. The only alternatives for weed management are cultural methods such as uprooting or interculture between rows, which raises production costs. No chemical control for broadleaf weeds is currently available, although chemical control for narrow leaf weeds is available in the form of selective weedicides.

### 5.3. Photoperiod Sensitivity and Heat Tolerance

The quinoa is cultivated as spring crop during November and harvested in April and May. Early crop growth stages usually enjoy low temperature (12–22 °C) which later increases during the reproductive period. The delayed sown quinoa often experiences high temperature (>30 °C) during flowering and anthesis termed as “Terminal Heat Stress” [99]. High temperature above 35 °C during flowering and seed filling stages causes significant reductions in seed yields of quinoa [53] associated with reduced pollen viability and empty inflorescence. The delayed cultivation of quinoa has been shown to reduce shoot and root growth traits, seed and biological yields including harvest index [18,56]. In addition, late sowing crop takes more days to complete the true leaf, four leaves, multiple leaves, and bud formation stages [18]. Under open door plexiglass fitted canopies, with a light transmission index of about 0.8, quinoa plants exposed to terminal heat (±7 °C) during the anthesis stage reduced the panicle length and weight, 100-seed weight and seed yield per plant including above ground dry matter in quinoa genotype UAF-Q7 [53,54]. These reduced yields were attributed to a decrease in gas exchange attributes, photosynthetic pigments, and decline in enzyme activities of antioxidants’ defense system under terminal heat [55]. Delayed sowing of quinoa with terminal heat had also reduced seed nutritional quality [55]. Nonetheless, high temperature stress during flowering has been found to produce longer panicles and more branches with delayed maturity showing quinoa adapt avoidance mechanisms to heat (Personal observation). As quinoa is a photoperiod sensitive crop, its cultivation in new regions is influenced by day length [36]. This crop is also genotype specific, which may affect crop growth duration [13,14]. Therefore, to reduce the negative impact of terminal heat, good yielding cultivars with early to medium duration should be identified in Pakistan’s irrigated conditions.

### 5.4. Control of Plant Height and Lodging Resistance

The crop has been ignored for decades and only rudimentary genetic changes have been made until now. To achieve maximum potential of quinoa as a fully domesticated crop, attempts to develop the plant by breeding have been limited [100]. Cereal crops lodging resistance is mostly determined by plant height [101]. Quinoa plants can grow up to 3 m tall in South America, posing a threat for lodging [102]. Additionally, environmental conditions influence plant height in quinoa and several experiments have found a negative link between plant height and seed yields for certain cultivars [102]. Under Faisalabad, Punjab, Pakistan condition, quinoa gain undesirable height (more than 120 cm) when day length and temperature start increasing after mid-February [17]. This result might be due to the amaranth form nature of adaptable genotypes that leads to lodging and stem breakage if a storm prevails. It may also be due to the hollow nature of the stem in quinoa [3]. Studies are in progress to use gibberellic acid inhibitor to control height and to avoid lodging and stem breakage issue in quinoa cultivars. Besides that, phenotyping studies are in progress to identify short stature genotypes from germplasm collection obtained from various countries. 

To avoid lodging without detrimental effects on quinoa yields, efforts should focus on genes that influence plant height. The quinoa genome includes two homologues of wheat Rht-B1/Rht-D1 (AUR62039523 and AUR62014191), which are both homologues of Arabidopsis RGA1 and encode a transcription factor involved in gibberellin signal transduction [103]. In comparison, no direct homologue of the GA20ox2 gene has been discovered [100].

### 5.5. Grain Number/Size and Yield Stability

Grain size is a desirable trait of crop improvement and consumer preference. Quinoa grains from southern highlands of Bolivia are of larger size than other ecotypes and affected by temperature during grain filling in this region due to photoperiod sensitivity of these ecotypes [104]. However, it should be noted that grain weight is more strongly affected by the grain filling rate [105,106] than by the duration of grain filling [107]. On the other hand, grain growth rate is negatively affected by high temperature and longer photoperiods; therefore, it is possible to select larger grains through breeding without affecting the duration of the grain growth under the irrigated conditions of Pakistan for genotypes of early to medium duration. 

Similarly, selection for seed yield and seed sizes can be achieved simultaneously as both traits are independent of genetics and environment (G × E) interaction among quinoa cultivars [104]. As yield from a farm scale is rarely reported, yield data obtained across different environments of approved cultivar can be used to establish a reference point or baseline to begin improvement for yield. The average yield potential (1500–2000 kg ha^−1^) of recently introduced quinoa cultivar UAF-Q7 in Pakistan can be utilized as a baseline for further quinoa genotype selection and yield enhancement under irrigated conditions. Even so, gains in quinoa yield should not be achieved at the expense of decreased nutritional and end-use quality.

Several other traits such as leaf area, total chlorophyll, number of branches, dry weight, and inflorescence per plant including harvest index have a positive association with seed yield. Hence, they can also be used as indirect selection traits in the yield improvement of quinoa crop [67,106].

### 5.6. Molecular Breeding and Genetic Approaches for Traits Improvement

Several breeding methods such as hybridization, interspecific crosses including simple, and reciprocal and passive crossing are carried out in quinoa to recombine desirable traits found in different species to next generation and for significant variation under abiotic conditions [108]. Individual and mass selection are applied for seed multiplication of quinoa cultivars developed from landraces to preserve their identity and composition of established cultivars while mutagenesis has been employed for improvement in plant type for vigor, yield potential, and decrease in saponin contents in quinoa [108,109]. In Pakistan, currently, information on quinoa breeding is scanty.; However, the selection of genotypes based on their adaptability, yield performance and low saponin contents is in progress. Conversely, considering the challenges of early vigor, seed size, yield stability, lodging resistance, heat tolerance, and low saponin grain contents, individual and mass selection and mutagenesis breeding techniques can be of potential application to develop a sustainable breeding program in Pakistan. 

Recently, Jarvis et al. [110] published high-quality genome data for quinoa which has opened new avenues for using targeted genome editing for evaluating adoption of this crop into new geographical areas different from its origin such as Pakistan and improving its agronomic performance. 

As an allotetraploid species, novel genome-editing technologies, such as CRISPR can be used efficiently to develop new varieties with reduced plant height to improve lodging resistance and knock out genes of saponin contents to produce sweet quinoa. However, this would require regulation through GM legislation before commercialization. Alternatively, technologies such as high-end TILLING as molecular breeding tool can be applied to speed up the varietal development program [100].

Marker assisted selection (MAS) following the identification of quantitative traits loci (QTLs) for increasing seed size and grain number and combining them in cultivar with similar genetic background can be a potential target for improving seed yield in quinoa. For this, two close homologues AtCKX5 and another two of AtCKX3 have been mapped in quinoa genome [100,110]. Likely, a two-gene sequence associated with saponin production has been identified that needs to be repressed in advanced generations to produce saponin free quinoa varieties. This will reduce 30% of the costs associated with quinoa production [98,110].

### 5.7. Socio-Economic Constraints and Adaptability in Existing Cropping Patterns

Quinoa adapts easily to existing technology practices followed in a region and responds by expressing all traits for better agronomic performance of a highly productive crop. Since the declaration of International Year of Quinoa in 2013 by the United Nations, with increasing demand, crop cultivation has been expanded in more than 120 countries and commercially produced outside Andean regions including US, Canada, India and China [22,111,112]. Even in the Middle East and North Africa Region (MENA) and the European Mediterranean regions, the crop has been successfully cultivated in marginal environments on salt affected soils and when using salt water for irrigation [22,113,114,115,116]. Furthermore, its adaptation to these environments is based on experimental results and several genotypes have been in the process of selection and approval [113,117,118,119]. 

In Pakistan, since the release of the first quinoa cultivar UAF-Q7 in 2019 for commercial cultivation, the crop has been cultivated on more than 200 ha. Several progressive growers, food companies, and retailers are involved in selling to local and international markets as quinoa grain and value-added products. 

If quinoa cultivation sees a further expansion in Pakistan, it will become a major crop. On the other hand, it may also stagnate if market demand decreases, or consumer demand fluctuates. Pakistan needs to adjust its quinoa production to market-driven demand of both local and international markets and has to meet internationally agreed quality standards to be able to compete with other stakeholders in the region including MENA, China, and India. The ball of contention between small to medium quinoa growers is the market access and quality maintenance for sustaining a profitable business. Now it is not merely a nutritional concern, as daily requirement can be fulfilled via quinoa’s replacement to wheat and rice. However, studies on quantitative comparisons are missing. Additionally, information on phytoremediation potential is missing in case of this notorious halophyte. There is still a considerable lack of research concerning biomass production of quinoa and its value as forage. Further, growing quinoa in existing cropping systems will compete with major crops for cultivated areas such as wheat or oilseeds or we have to cultivate the crop in small areas in rotation with other crops, such as rice–wheat or cotton–wheat. On other hand, we are also facing challenges of urbanization by bringing more cultivated areas under housing schemes. Pakistan spends millions of USD on oilseed import to meet vegetable oil requirements. It is still a burning question whether producing quinoa will reduce the burden of imports or not. Furthermore, growing quinoa on cultivated land under irrigated conditions as a low input crop with less fertilizers may degrade the soil even more. Nonetheless, increased quinoa production in Pakistan will raise concerns about its long-term sustainability as compared to the Andean area, where average yields of 600 kg ha^−1^ may lag if prices increase or decrease in the long run [22].

## 6. Conclusions and Future Prospects

Quinoa is famous due to its extraordinary nutritional profile, climate resilience, and extreme adaptability to adverse climates. Thus, it is the most potential crop that can ensure future global food and nutritional security in the developing countries. Despite quinoa expansion in more than 120 countries worldwide, the quinoa cultivation in Pakistan is still in experimentation since its introduction in 2009. The first commercial variety UAF-Q7 was released in 2019, and it is being cultivated throughout the country. There is lack of a breeding program for germplasm improvement regarding superior features of quinoa such as high yield and adaptability in different agro-ecological conditions. Access to more quinoa germplasm for maximizing genetic diversity is needed. There is lack of awareness about the nutritional and health benefits of quinoa among consumers and the unstructured market for farmers are major challenges in the promotion of crop. The relatively low productivity of existing quinoa variety, lack of quality seed, undesirable traits, and high market prices compared to other crops restricts its further scaling in Pakistan. 

Quinoa requires continued promotion until it becomes a part of the main food chain of common people. The role of policy makers, research institutions, farmers, and supply chain are important for its production and consumption. It is very important to vitalize the market and promote its consumption which in turn will trigger the increasing demand for quinoa production among smallholder farmers. Unfortunately, there is limited development in quinoa related products. In addition to the local market, the international market should be explored for export of high-quality quinoa grains matching the consumer demand with good branding. 

Candidate lines with low saponin content and grains of bold size could be helpful for marketing purposes and reducing production costs [95]. The high yielding quinoa varieties with wide adaptability under various agroecological zones are required. Pakistan needs to develop organic certification bodies for achieving maximum returns from this crop in the global market. Postharvest operations for saponin removal are complicated and need investment in mechanization to reduce the procedure. Mechanization in quinoa cultivation due to troublesome weed pressure need identification of cultivars with herbicide tolerance and of early vigor to reduce crop-weed competition. For this purpose, it is desired to introduce low-cost machinery for production and processing of quinoa among the growers and industrialists. As a spring crop, heat stress during the reproductive period due to increasing temperature is challenging to reduce detrimental effects on yield. Given the high protein content in the vegetative parts of quinoa, varieties with high biomass and productivity can be of particular interest as a nutritious fodder for livestock. Germplasm enhancement efforts through pre-breeding, quantitative and participatory breeding, as well as marker assisted selection for potential traits such as grain yield, high biomass, less saponin, and pollen viability need to be explored in adaptable quinoa germplasm. The successful development of quinoa value chains in Morocco offers a perspective to improve food and nutritional diversity of quinoa in Pakistan in a similar way [98].

## Figures and Tables

**Table 1 plants-11-01603-t001:** Description of phenological stages of quinoa accession under agro-climatic conditions of Faisalabad-Pakistan during 2020–2021.

Sr #	Description of Stage	Days after Sowing	Image
1	Emergence of Cotyledons	4–5	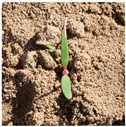
2	Emergence of true leaves	17–19	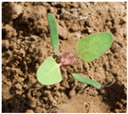
3	Visible bud appearance	65–68	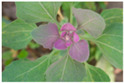
4	Anthesis	83–85	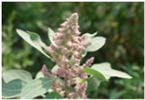
5	Physiological maturity	125–126	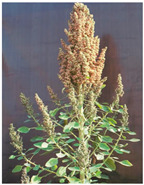
6	Harvest maturity	153–168	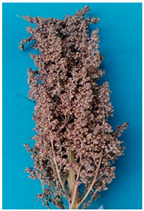

**Table 2 plants-11-01603-t002:** Comparative proximate analysis of UAF-Q7 quinoa cultivar based on published reports [7,33,69,70,71].

%	[33]	[7]	[69]	[70]	[71]
Ash	2.44	3.80	3.20	3.00	3.70
Protein	13.47	16.50	16.70	15.60	12.50
Fat	5.59	6.30	5.50	7.40	8.50
Fiber	2.71	3.80	10.50	2.90	1.90

Note: [33] values are average of four genotypes.

**Table 3 plants-11-01603-t003:** Comparative mineral analysis of quinoa grains based on published reports [7,8,33,72].

Minerals (mg/kg)	[33]	[8]	[7]	[72]
Ca	691.00	940.00	1487.00	1020.00
Copper	4.49	37.00	51.00	ND
Iron	64.47	168.00	132.00	105.00
Potassium	8877.98	ND	9267.00	8225.00
Magnesium	2115.70	2700.00	2496.00	ND
Manganese	32.72	ND	ND	ND
Sodium	48.14	ND	ND	ND
Phosphorous	4523.55	1400.00	3837.00	1400.00
Sulphur	1549.06	ND	ND	ND
Zinc	28.67	48.00	44.00	ND

Ref. [33] values are average of four genotypes; ND = Not detected.

**Table 4 plants-11-01603-t004:** Companies marketing quinoa products in Pakistan. (Website accessed date is 8 June 2022).

Sr #	Company Name	Product	Price (USD/kg)	Website
1	Khalis Things	Whole grain washed quinoa	8.33	https://khalisthings.com
2	The Soul Food Company	Prewashed quinoa	9.61	https://getsoulfood.com
3	Amna’s	Organic quinoa	7.77	https://amnasorganics.com
4	Virsa agri farms	Quinoa grain, multigrain flour	5.50	https://virsaproducts.com.pk
5	Shazday	White quinoa flour	8.88	https://freshbasket.com.pk
6	Farm Fresh	White quinoa, multigrain flour	6.00	https://farmfresh.com.pk
7	Syed Flour Mills	Quinoa ka Dalya	10.00	www.tradekey.com.pk
8	Gold Tree Millers	White quinoa	5.50	https://goldtreemillers.com
9	Hunter Foods	White and tri-color quinoa	13.30	https://www.hunterfoods.com
10	Family Foods	Organic White Quinoa	4.44	info@familyfoodproducts.com.pk
11	One Organics	Whole grain quinoa	4.22	https://www.daraz.pk
12	Natures Hug	Tri-color quinoa	20.20	https://www.alfatah.pk
13	Morganic	White quinoa grain, multigrain flour	9.40	https://www.morganic.com.
14	Quill (Bin Hashim Pharmacy)	Quinoa grain	9.08	https://binhashimonline.pk
15	Nutricles	Multigrain flour	8.05	https://nutricles.com
16	Healthhut	Quinoa grain	8.80	https://www.healthhut.pk
17	Daali Earth Foods	Quinoa grain	8.50	https://www.daaliearthfoods.com.pk
18	Ashley Foods	Quinoa grain	8.05	https://www.ashleyfoodsinc.com
19	Natural Foods	Quinoa grain	9.30	https://naturals.pk
20	Meadows Organic	Organic white quinoa	9.00	http://meadows-glutenfree.com
21	Sarang Herbs and Food	Quinoa grain and flour	8.33	https://sarang.com.pk

## Data Availability

Not applicable.

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
