# Peer review of "Trends and Limits for Quinoa Production and Promotion in Pakistan"

_plants, 2022, doi:10.3390/plants11121603_

Round 1

Reviewer 1 Report

The manuscript describes potentials and problems related to quinoa production in Pakistan. It deals with different aspects of quinoa production, from cultivation to the quality and marketing of seeds.  The topics are interesting, but several critical issues are present.

The text is poorly organized, with a lot of information repeated in different Subsections and not in a coherent order, not very detailed in some parts, as for the breeding approaches for the genetic improvement of this crop and for those aimed at reducing the saponin content, very important for the promotion of quinoa in Pakistan and all countries.

A very extensive revision of English through a native speaker is necessary, in all Sections. In general, attention should be paid to the style of writing. There is often an inappropriate use of punctuation marks.

Only a few suggestions are given to the authors and listed below in detail, which I hope will be useful in light of a more extensive arrangement of the manuscript.

In general:

Some sentences need to be written correctly (including lines 62-63, 74-81, 178, 185-192, 159-160, 178, 185-192, lines 234-238, lines 245-247, lines 249-250, 280-287, lines 348-353, lines 362-367, lines 410-422, all 5.2 subsection, lines 427-431, lines 472-477, lines 512-514, lines 572-578 and 582-588).

There is a large use of ‘therefore’, ‘however’, ‘nonetheless’, ‘as’, which makes reading of the manuscript very heavy.

In some sentences references need to be added (lines 74-81, 87, 207, 216-221, 242-245, 392, Subsection 5.2).

There is often an incorrect way of citing references.

Along the whole text, at the beginning of the sentence do not write the reference number directly in the square brackets but first put the name of the authors, eg. at line 149: "Saleem et al. [24], or “In another study, Iqbal et al. [25] (lines 152) and at lines 162, 167, 173, 175, 182, 185, 234, 323, 352, and others that have escaped me.

Other references in the text must be corrected, such as at line 132: “Sosa‐Zuniga et al. [37]”, removing [37] at line 134. The same at lines 152 and 159, lines 281 and 283, lines 362 and 364, lines 368 and 369, lines 388 and 390, lines 410 and 414, lines 535 and 538, and others that have escaped me.

Subsection 3.3. should be moved before Subsection 3.2.

Subsection 3.6. should be moved before Subsection 3.5.

Subsections 5.1. - 5.5., can be suitably moved previously in the relevant Subsections (3.2, 3.3), and appropriately integrated into them.

In particular:

Abstract - Line 21: farmers' association

Keywords - Line 24: “challenges; opportunities” are singular keywords. Others should be found that are more suitable and relevant

Line 34: “nutritional quality” should be more suitable

Line 39-40: “contents…content…contents”, is very repetitive and with some mistakes.

Line 44: “Its grains”

Line 48: “The quinoa plant life cycle…” start another paragraph

Line 52: “….however, it takes about 109-182 days in Europe”, write better

Line 53: “In Pakistan,…” star a new paragraph

Line 59: check references. Ref. 20 describes results of hydroponic culture and not of quinoa grown in South Punjab

Line 72: “Basic production technology”, do you mean "basic farming (cultivation) practices"?

Line 73: define “NPK” and write better

Lines 97-100: The sentences concern the UAF-Q7 genotype, described at the end of the previous Section which can be enriched with this information.

Lines 104-106: it is more correct “Generally, genotypes with thick stem, more branches and moderate plant height produce higher biomass for fodder, while genotypes with compact inflorescence are ideal for grain purpose”

Lines 114-117: “The center of quinoa diversity is the southern Andean highlands.” and “…germplasm center has been established at Andean region” are a repetition and these statements can be integrated.

Line 134. “The large grain size quinoa is preferred [3].”

Lines 135-136: nutritional quality is not expressed only by protein content. Deepen this topic

Line 142: replace “this crop” with “quinoa”

Line 166: “the tolerance and plasticity have been reported…”

Line 168: “while” is better

Line 171: delete “in quinoa”

Lines 171-172: this is a repetition and can be deleted

Lines 180-193: it might be more appropriate to move the effects of the high temperature after those of the salt (after line 164)

Line 195: “on a range of soils”

Line 254: “sensitivity of herbicides to Chenopodiumon”? usually, there is a sensitivity of weeds to herbicides

Line 261: “…. can be a sustainable weed management ...”

Lines 266-269: useless, can be deleted

Line 277: “ protocols ….have been established”

Lines 290 and 301: “Faisalabad (Pakistan)”

Line 306-307: what do you mean by "stand establishment" upon sowing?

Line 344: “Temperature affects…” could be better

Subsection 3.5. The text is repetitive. Rewrite in the simplest and most coherent way

Lines 355-361: very detailed and rich in data. If you add the corresponding reference, you could summarize this data well, giving the more significant information

Line 429: What do you mean by "processing practices" in this context?

Line 435-444: This part must be described in another Section (perhaps 4. because it is necessary for products that may be more pleasing to consumers) and adequately deepened

Lines 475-477 are not related to photoperiod sensitivity

Line 481: do you mean "to improve the plant by breeding"?

Line 498: “discovered [97].”

Subsection 5.6. should be deepened.

Lines 551-552: really, saponin free varieties could reduce the costs of processing quinoa into end- products suitable for consumption rather than the costs of its production

Lines 611-613: candidate lines with low saponin and bold grain could be useful both for marketing and to reduce production costs

Line 629: “Rehamna (Morocco)” or “Rehamna, in Morocco,”

Table 1: does the table show data relating to a single accession in a single field trial? if not, the days after sowing should be shown as a range. However, in general, it is more correct to show a range. Furthermore, mention the source

Table 2 and 3: standardize the Titles, the numbers, and the text in the last rows. If in Tables 3 ND means ‘Not detected’, the continue lines ------ what do they correspond to? not analyzed? Specify. What are the units of measurement in Table 3?

Table 4: standardize the Table rows

Author Response

The Editor

Plants, MDPI

We are thankful to you and anonymous reviewer for their useful comments which are incorporated, and manuscript has been revised with highlights in color and improved. All authors have critically read the manuscript carefully for improvement of language. Hopefully the revise manuscript will be considered for further processing.

Yours sincerely

Dr Irfan Afzal

Comments

Response

Some sentences need to be written correctly (including lines 62-63, 74-81, 178, 185-192, 159-160, 178, 185-192, lines 234-238, lines 245-247, lines 249-250, 280-287, lines 348-353, lines 362-367, lines 410-422, all 5.2 subsection, lines 427-431, lines 472-477, lines 512-514, lines 572-578 and 582-588).

Thank you, all suggested changes have been incorporated and corrected.

There is a large use of ‘therefore’, ‘however’, ‘nonetheless’, ‘as’, which makes reading of the manuscript very heavy.

Corrected and replaced.

In some sentences references need to be added (lines 74-81, 87, 207, 216-221, 242-245, 392, Subsection 5.2).

Incorporated

There is often an incorrect way of citing references.

Corrected and incorporated.

Along the whole text, at the beginning of the sentence do not write the reference number directly in the square brackets but first put the name of the authors, eg. at line 149: "Saleem et al. [24], or “In another study, Iqbal et al. [25] (lines 152) and at lines 162, 167, 173, 175, 182, 185, 234, 323, 352, and others that have escaped me.

Corrected and incorporated

Other references in the text must be corrected, such as at line 132: “Sosa‐Zuniga et al. [37]”, removing [37] at line 134. The same at lines 152 and 159, lines 281 and 283, lines 362 and 364, lines 368 and 369, lines 388 and 390, lines 410 and 414, lines 535 and 538, and others that have escaped me.

Corrected and incorporated

Subsection 3.3. should be moved before Subsection 3.2.

Incorporated

 Subsection 3.6. should be moved before Subsection 3.5.

Incorporated

Subsections 5.1. - 5.5., can be suitably moved previously in the relevant Subsections (3.2, 3.3), and appropriately integrated into them.

Thanks for valuable suggestion

Abstract - Line 21: farmers' association

Incorporated  

Keywords - Line 24: “challenges; opportunities” are singular keywords. Others should be found that are more suitable and relevant

Revised and Incorporated

Line 34: “nutritional quality” should be more suitable

Revised and incorporated

Line 39-40: “contents…content…contents”, is very repetitive and with some mistakes.

Corrected and incorporated  

Line 44: “Its grains”

Corrected

Line 48: “The quinoa plant life cycle…” start another paragraph

Accomplished

Line 52: “….however, it takes about 109-182 days in Europe”, write better

Modified

Line 53: “In Pakistan,…” star a new paragraph

Modified

Line 59: check references. Ref. 20 describes results of hydroponic culture and not of quinoa grown in South Punjab

Incorporated

Line 72: “Basic production technology”, do you mean "basic farming (cultivation) practices"?

yes

Line 73: define “NPK” and write better

Incorporated

Lines 97-100: The sentences concern the UAF-Q7 genotype, described at the end of the previous Section which can be enriched with this information.

Modified

Lines 104-106: it is more correct “Generally, genotypes with thick stem, more branches and moderate plant height produce higher biomass for fodder, while genotypes with compact inflorescence are ideal for grain purpose”

Corrected

Lines 114-117: “The center of quinoa diversity is the southern Andean highlands.” and “…germplasm center has been established at Andean region” are a repetition and these statements can be integrated.

Incorporated

Line 134. “The large grain size quinoa is preferred [3].”

Lines 135-136: nutritional quality is not expressed only by protein content. Deepen this topic

Incorporated  

Line 142: replace “this crop” with “quinoa”

Removed

Line 166: “the tolerance and plasticity have been reported…”

Revised and corrected

Line 168: “while” is better

Incorporated

Line 171: delete “in quinoa”

Deleted

Lines 171-172: this is a repetition and can be deleted

Deleted

Lines 180-193: it might be more appropriate to move the effects of the high temperature after those of the salt (after line 164)

Moved

Line 195: “on a range of soils”

changed

Table 1: does the table show data relating to a single accession in a single field trial? if not, the days after sowing should be shown as a range. However, in general, it is more correct to show a range. Furthermore, mention the source

This table shows phenological data of single accession for two years (2020-21). The range of parameters for two years are added. The data is not published yet.   

Table 2 and 3: standardize the Titles, the numbers, and the text in the last rows. If in Tables 3 ND means ‘Not detected’, the continue lines ------ what do they correspond to? not analyzed? Specify. What are the units of measurement in Table 3?

Table 2 and 3 are corrected accordingly

Table 4: standardize the Table rows

Incorporated

The Editor

Plants, MDPI

We are thankful to you and anonymous reviewer for their useful comments which are incorporated, and manuscript has been revised with highlights in color and improved. All authors have critically read the manuscript carefully for improvement of language. Hopefully the revise manuscript will be considered for further processing.

Yours sincerely

Dr Irfan Afzal

Comments

Response

Some sentences need to be written correctly (including lines 62-63, 74-81, 178, 185-192, 159-160, 178, 185-192, lines 234-238, lines 245-247, lines 249-250, 280-287, lines 348-353, lines 362-367, lines 410-422, all 5.2 subsection, lines 427-431, lines 472-477, lines 512-514, lines 572-578 and 582-588).

Thank you, all suggested changes have been incorporated and corrected.

There is a large use of ‘therefore’, ‘however’, ‘nonetheless’, ‘as’, which makes reading of the manuscript very heavy.

Corrected and replaced.

In some sentences references need to be added (lines 74-81, 87, 207, 216-221, 242-245, 392, Subsection 5.2).

Incorporated

There is often an incorrect way of citing references.

Corrected and incorporated.

Along the whole text, at the beginning of the sentence do not write the reference number directly in the square brackets but first put the name of the authors, eg. at line 149: "Saleem et al. [24], or “In another study, Iqbal et al. [25] (lines 152) and at lines 162, 167, 173, 175, 182, 185, 234, 323, 352, and others that have escaped me.

Corrected and incorporated

Other references in the text must be corrected, such as at line 132: “Sosa‐Zuniga et al. [37]”, removing [37] at line 134. The same at lines 152 and 159, lines 281 and 283, lines 362 and 364, lines 368 and 369, lines 388 and 390, lines 410 and 414, lines 535 and 538, and others that have escaped me.

Corrected and incorporated

Subsection 3.3. should be moved before Subsection 3.2.

Incorporated

 Subsection 3.6. should be moved before Subsection 3.5.

Incorporated

Subsections 5.1. - 5.5., can be suitably moved previously in the relevant Subsections (3.2, 3.3), and appropriately integrated into them.

Thanks for valuable suggestion

Abstract - Line 21: farmers' association

Incorporated  

Keywords - Line 24: “challenges; opportunities” are singular keywords. Others should be found that are more suitable and relevant

Revised and Incorporated

Line 34: “nutritional quality” should be more suitable

Revised and incorporated

Line 39-40: “contents…content…contents”, is very repetitive and with some mistakes.

Corrected and incorporated  

Line 44: “Its grains”

Corrected

Line 48: “The quinoa plant life cycle…” start another paragraph

Accomplished

Line 52: “….however, it takes about 109-182 days in Europe”, write better

Modified

Line 53: “In Pakistan,…” star a new paragraph

Modified

Line 59: check references. Ref. 20 describes results of hydroponic culture and not of quinoa grown in South Punjab

Incorporated

Line 72: “Basic production technology”, do you mean "basic farming (cultivation) practices"?

yes

Line 73: define “NPK” and write better

Incorporated

Lines 97-100: The sentences concern the UAF-Q7 genotype, described at the end of the previous Section which can be enriched with this information.

Modified

Lines 104-106: it is more correct “Generally, genotypes with thick stem, more branches and moderate plant height produce higher biomass for fodder, while genotypes with compact inflorescence are ideal for grain purpose”

Corrected

Lines 114-117: “The center of quinoa diversity is the southern Andean highlands.” and “…germplasm center has been established at Andean region” are a repetition and these statements can be integrated.

Incorporated

Line 134. “The large grain size quinoa is preferred [3].”

Lines 135-136: nutritional quality is not expressed only by protein content. Deepen this topic

Incorporated  

Line 142: replace “this crop” with “quinoa”

Removed

Line 166: “the tolerance and plasticity have been reported…”

Revised and corrected

Line 168: “while” is better

Incorporated

Line 171: delete “in quinoa”

Deleted

Lines 171-172: this is a repetition and can be deleted

Deleted

Lines 180-193: it might be more appropriate to move the effects of the high temperature after those of the salt (after line 164)

Moved

Line 195: “on a range of soils”

changed

Table 1: does the table show data relating to a single accession in a single field trial? if not, the days after sowing should be shown as a range. However, in general, it is more correct to show a range. Furthermore, mention the source

This table shows phenological data of single accession for two years (2020-21). The range of parameters for two years are added. The data is not published yet.   

Table 2 and 3: standardize the Titles, the numbers, and the text in the last rows. If in Tables 3 ND means ‘Not detected’, the continue lines ------ what do they correspond to? not analyzed? Specify. What are the units of measurement in Table 3?

Table 2 and 3 are corrected accordingly

Table 4: standardize the Table rows

Incorporated

Reviewer 2 Report

I have no comments. 

Author Response

We are highly thankful to the valuable feedback of the reviewer. 

Round 2

Reviewer 1 Report

The manuscript is much improved but there are still fewer critical issues.

My suggestions between line 195 and Table 1 hey are not present among those reported in the authors' reply, although in some cases I find the corresponding changes. I don't understand how it happened, whether it was my mistake in their insertion or an oversight by you. However, in this report I have added those still present after the revision of the manuscript. In some cases (lines 59, 71, 182-183) the requested changes have not been made and, if it was not considered necessary to make them, the reason was not explained.

References to lines 74-76, 171-172, 375-377 are still missing.

Line 59: As already mentioned, reference 20 refers to the results in hydroponics, therefore in the laboratory, and not in the field in South Punjab. Either remove the reference or change the sentence, keeping to the general, for example "soils with electical conductivity .... like those of South Punjab".

Line 70: If you mean "Basic farming practices” write like this, as “basic production technology” is not the most suitable definition.

Line 71: “NPK” has not been specified. Usually, NPK are fertilizers composed of three elements, namely nitrogen, phosphorus and potassium. You can use these ternary fertilizers directly, but also the single elements in different concentrations.  When you talk about "NPK requirements", I guess you are referring to the requirement of the individual elements. So, explain this well.

Lines 94-95: “Generally, genotypes with thick stem, more branches and moderate plant height produce higher biomass and are ideal for fodder and genotypes with compact inflorescence are ideal for grain pourpose” or “Generally, genotypes with thick stem, more branches and moderate plant height producing higher biomass are ideal for fodder and genotypes with compact inflorescence are ideal for grain pourpose”.

Line 206: be careful to give the correct references. Ref 34 is not Saleem et al. but Iqbal et al.

Lines 182-183: as already mentioned, usually, there is a sensitivity of weeds to herbicides.

Line 189 “…. can be a sustainable weed management ...”.

Lines 226-227: it is the opposite.

Lines 228 and 274: “, in Pakistan conditions,”.

Line 230: “despite the fact that quinoa”

Line 355: Do you mean “Temperature affects…” or “Temperature increases the rate of deterioration in the presence of moisture content and humidity”?

Lines 317-318:  what do you mean by "stand establishment" upon sowing?

Line 262: “ protocols ….have been established”

Line 309: “Table 2 & 3.”

Tables: Why are Tables 2 and 3 not inserted after the relevant subsection 3.5?

Line 394: be careful to give the correct references. The reference [88] is not correct. The correct one is [94].

Line 414: What do you mean by "processing practices" in this context?

Line 425: In this subsection you use the term "Mechanization" in general, going from agricultural practices to the use of mechanized systems at harvest and post-harvest stages, up to mechanical pearling as a first step, not a chemical one, to decrease the saponin content. You could change the title more appropriately.

 Line 447: what does "except for flowering with earlier" mean? Write correctly.

Lines 457-461: Rewrite more clearly.

Line 463: “This crop”

Line 481: “two homologues….and encode….”

Lne 483: “discovered [97]”

Line 494: define “GxE”

Lines 535-536: really, saponin free varieties could reduce the costs of processing quinoa into end-products suitable for consumption rather than the costs of its production.

Line 612: I meant ”… and Rehamna, in Morocco, can be a success story for lesson to be learned”.

Tables (Lines 358 and 360): Standardize the titles as:

Table 2. Comparative proximate analysis of UAF-Q7 quinoa cultivar based on previous published reports.

Table 3. Comparative mineral analysis of quinoa grains based on previous published reports.

In Table 2, for each reference, decide how many decimal digits to add and use those for all the data in the same column (for istance, for [7] and [84]).

In Table 3 correctly edit the data in each column.

Author Response

Dear Editor

I have revised the manuscript according to the reviewer's comments. Kindly find here point by point response to the reviewer's comments;

The manuscript is much improved but there are still fewer critical issues. 

My suggestions between line 195 and Table 1 hey are not present among those reported in the authors' reply, although in some cases I find the corresponding changes. I don't understand how it happened, whether it was my mistake in their insertion or an oversight by you. However, in this report I have added those still present after the revision of the manuscript. In some cases (lines 59, 71, 182-183) the requested changes have not been made and, if it was not considered necessary to make them, the reason was not explained.

References to lines 74-76, 171-172, 375-377 are still missing.

Response: The suggestions are incorporated. Missing references are added.

Line 59: As already mentioned, reference 20 refers to the results in hydroponics, therefore in the laboratory, and not in the field in South Punjab. Either remove the reference or change the sentence, keeping to the general, for example "soils with electical conductivity .... like those of South Punjab".

Response: The sentence is rephrased.

Line 70: If you mean "Basic farming practices” write like this, as “basic production technology” is not the most suitable definition.

Response: Replaced with “Basic farming practices”

Line 71: “NPK” has not been specified. Usually, NPK are fertilizers composed of three elements, namely nitrogen, phosphorus and potassium. You can use these ternary fertilizers directly, but also the single elements in different concentrations.  When you talk about "NPK requirements", I guess you are referring to the requirement of the individual elements. So, explain this well.

Response: Revised accordingly

Lines 94-95: “Generally, genotypes with thick stem, more branches and moderate plant height produce higher biomass and are ideal for fodder and genotypes with compact inflorescence are ideal for grain pourpose” or “Generally, genotypes with thick stem, more branches and moderate plant height producing higher biomass are ideal for fodder and genotypes with compact inflorescence are ideal for grain pourpose”.

Response: Replace with “Generally, genotypes with thick stem, more branches and moderate plant height producing higher biomass are ideal for fodder and genotypes with compact inflorescence are ideal for grain pourpose”.

Line 206: be careful to give the correct references. Ref 34 is not Saleem et al. but Iqbal et al.

Response: Corrected

Lines 182-183: as already mentioned, usually, there is a sensitivity of weeds to herbicides.

Response: Corrected

Line 189 “…. can be a sustainable weed management ...”.

Response: Changed

Lines 226-227: it is the opposite.

Response: Revised accordingly

Lines 228 and 274: “, in Pakistan conditions,”.

Response: corrected

Line 230: “despite the fact that quinoa”

Response: Revised

Line 355: Do you mean “Temperature affects…” or “Temperature increases the rate of deterioration in the presence of moisture content and humidity”?

Response: Temperature increases the rate of deterioration in the presence of moisture content and humidity. Revised now.

Lines 317-318:  what do you mean by "stand establishment" upon sowing?

Response: Revised

Line 262: “ protocols ….have been established”

Response: Corrected

Line 309: “Table 2 & 3.”

Tables: Why are Tables 2 and 3 not inserted after the relevant subsection 3.5?

Response: Inserted after the relevant subsection 3.5

Line 394: be careful to give the correct references. The reference [88] is not correct. The correct one is [94].

Response: Corrected this reference. All the references are crosschecked.

Line 414: What do you mean by "processing practices" in this context?

Response: Processing practices mean the removal of saponin contents which are anti-nutritional components in the quinoa seed coat.  

Line 425: In this subsection you use the term "Mechanization" in general, going from agricultural practices to the use of mechanized systems at harvest and post-harvest stages, up to mechanical pearling as a first step, not a chemical one, to decrease the saponin content. You could change the title more appropriately.

Response: Further detail is added. Mechanization is required from sowing to harvest of quinoa cultivation in developing countries.

 Line 447: what does "except for flowering with earlier" mean? Write correctly.

Response: Revised accordingly

Lines 457-461: Rewrite more clearly.

Response: Revised

Line 463: “This crop”

Response: Revised

Line 481: “two homologues….and encode….”

Response: Revised

Lne 483: “discovered [97]”

Response: Revised

Line 494: define “GxE”

Response: It was elaborated.

Lines 535-536: really, saponin free varieties could reduce the costs of processing quinoa into end-products suitable for consumption rather than the costs of its production.

Response: Yes

Line 612: I meant ”… and Rehamna, in Morocco, can be a success story for lesson to be learned”.

Response: Corrected

Tables (Lines 358 and 360): Standardize the titles as:

Table 2. Comparative proximate analysis of UAF-Q7 quinoa cultivar based on previous published reports.

Table 3. Comparative mineral analysis of quinoa grains based on previous published reports.

In Table 2, for each reference, decide how many decimal digits to add and use those for all the data in the same column (for istance, for [7] and [84]).

In Table 3 correctly edit the data in each column.

Response: Tables are corrected accordingly.
